# Research Progress and Potential Applications of Spermidine in Ocular Diseases

**DOI:** 10.3390/pharmaceutics14071500

**Published:** 2022-07-19

**Authors:** Wentao Han, Haoyu Li, Baihua Chen

**Affiliations:** 1Department of Ophthalmology, The Second Xiangya Hospital of Central South University, Changsha 410011, China; hanwentao1998@163.com (W.H.); med.dr.lihy@gmail.com (H.L.); 2Hunan Clinical Research Center of Ophthalmic Disease, Changsha 410011, China

**Keywords:** spermidine, ocular diseases, eye, biomarker, therapeutic agent

## Abstract

Spermidine, a natural polyamine, exists in almost all human tissues, exhibiting broad properties like anti-aging, autophagy induction, anti-inflammation, anti-oxidation, cell proliferation activation, and ion channel regulation. Considering that spermidine is already present in human nutrition, recent studies targeting supplementing exogenous sources of this polyamine appear feasible. The protective role of spermidine in various systems has been illuminated in the literature, while recent progress of spermidine administration in ocular diseases remains to be clarified. This study shows the current landscape of studies on spermidine and its potential to become a promising therapeutic agent to treat ocular diseases: glaucoma, optic nerve injury, age-related macular degeneration (AMD), cataracts, dry eye syndrome, and bacterial keratitis. It also has the potential to become a potent biomarker to predict keratoconus (KC), cataracts, uveitis, glaucoma, proliferative diabetic retinopathy (PDR), proliferative vitreoretinopathy (PVR), and retinopathy of prematurity (ROP). We also summarize the routes of administration and the effects of spermidine at different doses.

## 1. Introduction

Natural polyamines consist of spermine, spermidine, and putrescine [1]. Spermidine is well-known as an anti-aging agent and exerts several protective effects [2]. Numerous studies on human and animal models have shown it to delay the aging of the whole organism and extend the median lifespan [3,4]. In the central nervous system, spermidine improved cognitive function in diabetic murine and drosophila aging models [5,6] and ameliorated neuron injury in an ischemic preconditioned murine model [7]. In the cardiovascular system, it controls high blood pressure and prevents heart failure in murine and human beings [3,8,9]. In the immune system, it helps to maintain mucosal homeostasis of the intestine, which is under excessive inflammation [10,11,12]. It also facilitates the regeneration and migration of epithelial cells, thus protecting barrier function in some cases, such as retinal pigment epithelial cells (RPEs) and colon epithelial cells [10,13,14,15].

In the light of such protective effects, researchers have recently expanded its application to ocular diseases such as dry eye syndrome, bacterial keratitis, cataracts, glaucoma, optic nerve injury, and age-related macular degeneration (AMD) [14,16,17,18,19,20]. These diseases pose a significant threat to human vision worldwide, especially among working-aged groups, thus exerting a heavy burden on families and society [21,22]. Since eye health is becoming a growing concern, novel prevention and treatment methods are attracting more and more attention to restore vision and improve the quality of life. Although the role of spermidine in various systems has been illuminated and summarized in the literature [23], the recent progress of spermidine administration in ocular diseases remains to be clarified. These findings indicate that spermidine may exhibit great diagnostic and therapeutic potential for ocular diseases.

## 2. Metabolism of Spermidine

Spermidine [N-(3-aminopropyl)1,4-diaminobutane], a small molecule with a relative molecular weight of 145.25 Daltons, is abundant in a broad palette of plant and animal food [24]. The amount of spermidine in the human body is maintained through absorption of nutrients, intracellular biosynthesis, and microbial production in the gut [25,26]. Clinically, as a pharmacological caloric restriction mimetic (CRM), its low toxicity yet strong efficacy has been elucidated in clinical trials. Because of different dietary habits, there is regional variation in the recommended daily minimum of spermidine ranging from 5 to 15 mg [4].

Spermidine is distributed in almost all human tissue and can be synthesized endogenously. In the internal environment, it exhibits a net positive charge, so it cannot transport directly across the cell membrane; therefore, exogenous spermidine is taken up from the extracellular space through active transportation and endocytosis [27,28]. In contrast, endogenous spermidine can be synthesized via two principal pathways. One concerns the formation of putrescine (the precursor of spermidine) from L-arginine by two chemical reactions (Figure 1). In mammalian cells, putrescine can be synthesized directly via ornithine decarboxylation under catalysis from the rate-limiting enzyme ornithine decarboxylase (ODC). The other pathway is decarboxylation of arginine to agmatine by arginine decarboxylase (ADC), which is then converted into putrescine via the enzyme agmatinase [29,30,31,32]. Conversion between polyamines can also occur. For instance, spermine can gradually decompose into spermidine and putrescine under the catalysis of spermidine/spermine-N1-acetyltransferase (SSAT)—N1-acetylpolyamine oxidase (APAO) [33]; it can also be directly converted to spermidine under the action of spermine oxidase (SMO), but this process may generate harmful compounds such as hydrogen peroxide (H_2_O_2_) and acrolein, which could be toxic and induce oxidative stress in excessive accumulation [34,35,36]. Therefore, the spermidine level in vivo is determined by polyamines, including spermine and putrescine, predominantly initiated by the amino acids ornithine, methionine, and arginine.

## 3. Biofunctions of Spermidine

Spermidine has a wide range of biofunctions: anti-aging, autophagy induction, cell proliferation activation, anti-inflammation, anti-oxidation, and ion channel regulation [14,37,38,39,40,41,42,43,44,45].

Mechanically, aging enhances the formation of some proteins and organelles, which are regarded as damaged or unnecessary cytoplasmic components [46,47]. Aging cells cannot degrade them, while autophagy delivers these components into lysosomes, which function as a complex intracellular degradation system [48]. Therefore, autophagy induction has been recognized as an effective strategy to extend life, while autophagy deficiency may shorten it [37]. Spermidine, a well-known autophagy inducer, upregulates autophagy-related genes [23,49] and the expression of transcription factors (e.g., elF5A and TFEB [50]). Spermidine can also inhibit protein acetylation by reducing the expression of E1A-associated protein p300 (EP300), which promotes the directly acetylation of multiple autophagy-essential proteins and stimulates the deacetylation of tubulin by indirectly inhibiting a-tubulin acetyltransferase 1 (aTAT1) [51,52,53]. In addition, spermidine has been proven to be a precursor for the essential enzymatic modification of eIF5A [54]. It may also play a positive role in stem cell function, such as the proliferation and migration of epithelial cells or slow-proliferating primary cultures [13,17,39,55], by stimulating. them to differentiate and prevent their senescence [23], partly through promoting entry into the S/G2-M phases of the cell cycle [14,38]. Regarding anti-inflammation, spermidine downregulates proinflammatory cytokines such as IL-6, TNF-α, and IL-1β. It does this by inducing macrophage polarization and CD4^+^ T cell differentiation in an autophagy-dependent manner to modulate the amount and function of immune cells [40,41,56]. When cells or tissues are confronted with oxidative or endoplasmic reticulum stress, spermidine acts as an antioxidant as well as a specific [Ca^2+^]_I_ chelator that rescues cells from Ca^2+^ overload induced by oxidative stress thereby preventing apoptosis [14,39,42,57]. It significantly reduced oxidative markers and endoplasmic reticulum stress-related proteins in cell models that simulated bone joint and lung fibrosis in an autophagy-dependent manner [39,42]. The supplementation of spermidine also maintains mitochondrial homeostasis by upregulating the expression of sirtuin 1 (SIRT1), which deacetylates peroxisome proliferator-activated receptor γ coactivator-1 α (PGC-1α), a key regulator of mitochondrial biogenesis and function. As a result, the accumulation of ROS induced by H_2_O_2_ during the protection of cardiac aging is inhibited [43]. Additionally, spermidine has the potential for neuroprotection by modulating the ion channel to inhibit the N-methyl-D-aspartate receptor (NMDAr). It is a glutamate receptor that mediates glutamate’s excitatory role in the pathogenesis of some diseases of the central nervous system (e.g., epilepsy) because it acts on NR2B subunits of NMDAr to inhibit Ca^2+^ influx and excitotoxicity. NMDAr can also decrease inducible nitric oxide synthase (iNOS) and NO, which are triggered by NMDAr activation [44,45].

## 4. Role of Spermidine in Ocular Diseases

### 4.1. Role of Spermidine in Ocular Surface Disorders

#### 4.1.1. Dry Eye Syndrome

Recently, Lee et al. applied spermidine (0.2, 0.5%) in the form of topical eye drops in a rat model of dry eye syndrome induced by fine particulate matter (aerodynamic diameter below 2.5 µm, PM_2.5_) to simulate air pollution. The results showed that spermidine stabilized the tear film and suppressed ocular surface inflammation [17]. In that study, proinflammatory cytokines such as IL-6, TNF-α, and IL-17 in the lacrimal gland and corneal tissue were significantly reduced after topical administration of spermidine (20 µL of 0.2% or 0.5% spermidine, four times daily). In addition, cornea epithelial cells were prevented from falling off after spermidine administration [17].

Mechanically, anti-inflammatory action is one of the critical means for controlling dry eye syndrome [47]. M1/M2 macrophage polarization and regulatory T cell (Treg)/T helper type 17 (Th17) cell balance play a crucial role in modulating inflammation, especially the abovementioned proinflammatory cytokines [41,58,59]. In comparison, the anti-inflammation effect of spermidine modulated the amount and function of immune cells by inducing autophagy-dependent macrophage polarization and CD4^+^ T cell differentiation [40,41].

Two distinct phenotypes of polarized macrophage activation have been recognized: the classically activated (M1) and the alternatively activated (M2). Functionally, M1 macrophages secrete proinflammatory and T cell polarizing cytokines, inducing inflammation and tissue destruction, while M2 macrophages usually secrete anti-inflammatory cytokines, resolved in modulating inflammation and promoting wound healing [49]. As mentioned above, exogeneous spermidine is taken up from the extracellular space through active transportation and endocytosis [27,28], which may be the primary mechanism in mammals [60]. Macrophages, the main components of the mononuclear phagocyte system, are recognized as one of the most competent cells with endocytosis capacity [61], making them the primary target cells for administrated spermidine [62]. When administrated, it can induce macrophage polarization [63]. In the dry eye syndrome murine model induced by PM_2.5_, activation of M1 macrophages in the lacrimal gland and conjunctiva was suppressed by spermidine, effectively controlling excessive inflammation [21].

CD4^+^ T cells can differentiate into Treg and Th17 cells, and the Th17/Treg cell balance has proved crucial in regulating autoimmunity and tolerance [64]. Excessive Th17 cell polarization can stimulate the secretion of inflammatory cytokines such as interleukin 17 (IL-17); meanwhile, inflammation induced by macrophages promotes CD4^+^ T cell differentiation into Th17 cells, which in turn suppresses the function of Treg cells [65]. Altered Th17/Treg cell balance has been confirmed in ocular diseases such as Sjögren’s syndrome, dry eye syndrome, optic neuritis, and Posner–Schlossman syndrome [17,66,67,68,69]. In dry eye syndrome caused by PM_2.5_, lowered IL-17 was seen in the cornea and lacrimal gland in the experimental groups, indicating that spermidine modulates CD4^+^ T cell differentiation [17]. Unfortunately, researchers have not yet traced the root of spermidine to reveal the mechanism by which it modulates the inflammatory status and retrieves impaired tissue structures from the ocular surface [17].

#### 4.1.2. Bacterial Keratitis

*Staphylococcus aureus* (*S. aureus*) is a leading cause of bacterial keratitis, which is associated with severe inflammation, serious ocular damage, and permanent vision loss [70]. Clinically, the most effective treatment is broad-spectrum antibiotics. However, with the abuse of antibiotics, methicillin-resistant *S. aureus* (MRSA) is a growing concern, where broad-spectrum antibiotic treatments have limited effect [71]. Meanwhile, spermidine is known to exert a strong synergistic effect with β-lactams against MRSA [72]. When applied in combination with β-lactam antibiotics, the minimum inhibitory concentrations (MICs) of antibiotics are significantly reduced, which may be explained by increased penicillin-binding proteins (PBPs), acylation by β-lactam antibiotics, or decreased expression of PBPs caused by spermidine [73]. Therefore, spermidine exhibited strong antibiotic and harmlessly transient tight-junction modulating effects on cornea epithelial barriers. It noticeably ameliorated bacterial keratitis when converted into carbon quantum dots (CQDs), a nanomaterial with antimicrobial efficacy, through simple dry heating at 260 °C [18]. Topical administration of the nanomaterial exerted remarkabe antibacterial effects on non-multidrug-resistant bacteria: *Escherichia coli, S. aureus, Pseudomonas aeruginosa,* and *Salmonella enterica serovar Enteritidis*) and MRSA due to its strong disintegrating effect on the bacterial membrane [18]. Only a tiny portion of polyamines are combined onto bacterial membranes: less than 1% covalently bind to bacterial membrane peptidoglycans. At the same time, CQD–spermidine exhibits a strong multivalent interaction with bacterial membranes due to its much larger surface curvature, which contains a high density of spermidine [18].

On the other hand, the strong, positively charged CQDs-spermidine could bind to DNA or siRNA altering their conformation. This would prevent DNA duplication and gene expression and lead to disturbed bacteria cell function [18]. Therefore, the synergistic destabilization of cell membranes and inhibition of membrane synthesis resulting from the binding of CQDs-spermidine may account for a 2500-fold lower MIC than for spermidine alone [18]. It is also noteworthy that spermidine promotes the function of the gut microbiota in inflammatory bowel diseases [74] even though the gut microbiota is supposed to be associated with keratitis [75].

However, some different voices emphasize that the application of spermidine alone may worsen bacterial infections. For instance, Hume et al. developed a keratitis model with spermidine treatment and an *S. aureus* inoculation, and it indicated that spermidine may positively affect *S. aureus* growth and virulence [76]. It may also hamper the antibiotic effect of phospholipase A(2) in tears when infected with *S. aureus* [77]. Therefore, when applied in the absence of antibiotics, the effect of spermidine in bacterial keratitis remains to be further investigated.

#### 4.1.3. Keratoconus (KC)

KC is an ectatic corneal disease marked by the protrusion of the frontal region of the eye caused by the thinning of the central corneal apex, which may lead to a significant reduction in visual acuity [78]. The defects in the extracellular matrix (ECM) deposition of KC-derived corneal fibroblasts (HKCs) play a decisive role in the KC pathology [79].

Mckay et al. cultured HKCs in vitro, revealing that levels of arginine and spermidine were significantly decreased compared with normal cells (7.5-fold lower, *p* = 0.03) [80]. After supplementation with arginine, collagen type I secretion of HKCs increased significantly. In contrast, the level of spermidine only increased steadily in normal corneal fibroblast cells [80], so the role of spermidine in KC remains to be elucidated.

### 4.2. Role of Spermidine in Cataracts

The lens’s transformation and synthesis of polyamines are restricted due to the minimal activities of ODC and S-adenosylmethionine decarboxylase. Therefore, lens polyamines (including spermidine) should be supplemented from the surrounding intraocular fluid [81]. The level of spermidine can gradually decrease with age, and some researchers assume that decreased spermidine in the lens may to some extent be associated with age-related cataracts [82]. In a murine model of traumatic cataracts, the level of lens spermidine decreased slightly after needle punctuation due to increased SSAT activity [83]. In the Ca^2+^-induced cataract model, simulating age-related cataracts, endogenous spermidine decreased rapidly as lens opacification progressed, in parallel with the increase in crystallin cross-linking by N1–N8-bis(c-glutamyl)SPD [19]. When incubating a rabbit lens in a high concentration of free (unbound) Ca^2+^ with exogenous spermidine in vitro, delayed lens opacification was observed in two studies [19,84], indicating that spermidine may become a promising agent for preventing cataracts, especially age-related, the most common subtype.

In cataracts, an increase in the intracellular levels of free (unbound) Ca^2+^ has been seen in the lens during aging, which may trigger enzymes, such as Ca^2+^-dependent transglutaminase2 (TG2) and decrease the activity of flavin adenine dinucleotide-dependent polyamine oxidases (FAD-PAO) [84]. In that circumstance, activated TG2 catalyzes lens crystallins cross-linked by protein-bound N1–N8-bis(γ-glutamyl)SPD. Meanwhile, the suppressed FAD-PAO does not degrade the reactive N8-terminal residue of the crystallin-bound N1-mono(γ-glutamyl)SPD, which reduces the crosslink formation, thus leading to crystallin polymerization and lens opacification [19,85,86]. Supplementation of spermidine can decrease TG2 activity by saturating its binding sites and enhancing the activity of FAD-PAO by more than double in vitro [84]. Administration of exogenous spermidine may restore the endogenous spermidine and reduce the N1–N8-bis(γ-glutamyl)SPD cross-linking while increasing mono(γ-glutamyl)SPD, which may be responsible for inhibiting the cross-linking of lens crystallins and promoting lens transparency [19].

### 4.3. Role of Spermidine in Uveitis

Uveitis is an inner-eye disease where inflammation could lead to permanent destruction of the neuroretina [87].

Spermidine has been proposed to be involved in inflammatory reactions, and its level in the aqueous humor was measured in a rabbit model of endotoxin-induced uveitis [88,89]. Wickström et al. disclosed that, unlike other polyamines (spermine and putrescine), the spermidine level decreased in the first 24 h after endotoxin injection and then rose, while spermine and putrescine were significantly elevated 24 h after endotoxin treatment [88]. An analysis of spermidine, spermine, putrescine, and acetylated spermidine in the aqueous humor, revealed correlations between polymorphonuclear leukocytes and putrescine, mononuclear leukocytes and spermine, and acetylated spermidine and lymphocytes [89]. In another study, they investigated polyamines of rabbit aqueous humor after surgical trauma [90]. In that research, spermidine decreased slightly on the first day and then rose significantly; the results of the other two polyamines were also consistent with the rabbit model of endotoxin-induced uveitis mentioned above. However, despite the significant increase of polyamines after surgical trauma, no correlation was found between the number of leukocytes and high levels of polyamines [90].

### 4.4. Role of Spermidine in Ocular Neuroprotection and Retinal Disorders

#### 4.4.1. Glaucoma, Optic Nerve Injury, and Retinal Nerve Degeneration

Glaucoma is a group of chronic optic neuropathies characterized by a defect of the visual field, progressive loss of retinal ganglion cells (RGCs), and cupped optic neuropathy [91]. It can be divided into specific subtypes each having different pathophysiological mechanisms: primary open-angle glaucoma (POAG), primary angle-closure glaucoma (PACG), and secondary exfoliation glaucoma (XFG) [92].

The metabolic signature of polyamines in aqueous humor and plasma samples from patients diagnosed with glaucoma were analyzed, revealing that decreased spermidine concentration may be associated with POAG and secondary open-angle glaucoma [93,94,95]. In a univariate analysis, the fold change of spermidine (POAG/control) reached a plasma metabolomic signature of 0.9 (*p* = 0.0017). Spermidine was also identified as a metabolite with critical coefficients through the lasso method, which contributed to the establishment of logistic regression models for POAG prediction. It received good area under the receiving operator characteristic curve (AUROC) values (generally over 90%) [93]. In a group of patients, 50% with glaucoma, who had been diagnosed with exfoliative syndrome, a leading cause of secondary open-angle glaucoma, an orthogonal partial least squares discriminant analysis (OPLS-DA) revealed generally lower spermidine concentrations than for the control group (fold change = 0.89, VIP = 0.87, *p* = 0.014) [94]. In an aqueous humor metabolomic signature, the spermidine content was selected as changed metabolites (variable importance in the projection, VIP = 1.43; *p* = 0.19) in OPLS-DA, yielding a receiver operating characteristic curve (AUC) value of 0.69 for POAG prediction [95]. In addition, in the murine model of optic atrophy with OPA1 gene deficiency, similar to glaucoma, the metabolic signature revealed a decreased concentration of spermidine (VIP ≈ 1.4) in optic nerve samples [96,97]. Therefore, these findings indicate that decreased spermidine may to some extent be associated with glaucoma, making it a potentially promising biomarker.

Decreased spermidine metabolism may play an integral role in POAG, secondary open-angle glaucoma, and optic nerve atrophy [93,94,95,96]. Supplementations with exogenous spermidine to alleviate the pathological progress of glaucoma and protect the optic nerve were carried out [16,98,99], and disclosing their role may further illuminate the pathophysiology of glaucoma and even lead to new therapeutic options. Noro et al. added spermidine in daily drinking water (30 mM) to a mouse model of normal-tension glaucoma, optic nerve injury, and multiple sclerosis and revealed that spermidine could ameliorate visual impairment as well as promote RGC survival and optic nerve regeneration [16,98,99]. In the retinal neurodegeneration induced by monosodium glutamate (MSG), which is neurotoxic, spermidine was subcutaneously injected at a dose of 10 mg/kg and exerted complete protection on the whole retina when exposed to a lower dose of MSG, as well partial prevention to the inner retinal layers treated by a high dose of MSG [100]. In a murine model, optic nerve injury (crushed at 2 mm behind the eye), after 20 pmol/eye (20 μM, 10 μL), intravitreal injection of spermidine significantly promoted optic nerve regeneration, which had previously undergone a crush for 10 s after the dural sheath surrounding the optic nerve was incised [20]. In addition, in a PM_2.5_ exposed murine model, an application of spermidine (0.2 and 0.5%) eye drop alleviated retinal damage such as the loss of RGCs, and tissue destruction of ganglion cell layer (GCL) and the inner plexiform layer (IPL) [17]. In some in vitro studies, the neuroprotective effect of spermidine was also confirmed. For instance, it exhibited both neuroprotective and anti-oxidative stress effects at a dose of 4 mM in RGC incubation against H_2_O_2_ [98], and it promoted optic nerve regeneration in vitro at a concentration of 20 µM; however, such effect diminished at 100 and was lost at 500 µM [20].

The neuroprotective role of spermidine may be owing to some different mechanisms. For example, oxidative stress induced by various factors can damage the neurovascular unit, which may delay the neuro-regeneration of RGCs and the optic nerve [101,102]. Noro et al. observed decreased retina oxidative stress levels, particularly for GCL, in a murine glaucoma model through spermidine administration through drinking water at a dose of 30 mM per day. The model was established via excitatory amino acid carrier 1 (EAAC1) gene knockout, which alleviated visual impairment and retinal degeneration of the glaucoma-like phenotype by acting as an antioxidant [16]. They also hypothesized that spermidine rescues GCL by inhibiting NMDAr, the effect of which was consistent with their previous finding in NMDAr antagonist applications in glaucoma. It depended on the route of polyamine administration, the nature of the molecules used (spermidine, putrescine, or spermine), and the dose [16,103]. The human retina possesses specific polyamine binding sites/NR2B, subunits of the NMDAr, while spermidine is not uniformly distributed in the retina, which has an especially rich GCL, or in the inner half of the inner nuclear layer where NMDAr was known to exist [104,105]. However, there are different voices regarding this hypothesis under other conditions, such as different doses [106,107]. For instance, spermidine injected intravitreally at 50 nmol/eye (a much higher concentration than Deng et al. [20]), inhibited the neuroprotective effect of nafamostat (an NMDAr inhibitor) in a NMDA-induced (5 µL, 20 nmol/eye) retinal degeneration model [106]. It is worth mentioning that intravitreal injection of spermidine at a 30 nmol/eye dose (10 µL) already caused significant necrosis in retinal pigment epithelium (RPE) cells and apoptosis in photoreceptor cells [108]. In cell culture, spermidine acts on the NMDAr in a biphasic concentration-dependent manner. In sub-millimolar doses, it blocks the receptors, but at sub-micromolar levels, it activates the receptors [109]. The protective effects of Ifenprodil (another NMDAr inhibitor) against glutamate neurotoxicity were inhibited by spermidine at a dose of 100 μM in retinal neuronal cells [107].

In contrast, a 4 mM spermidine treatment for 16 h in an RGC culture significantly promoted cell viability in another in vitro study [98]. In a murine model of optic nerve injury (intraorbital exposed and crushed with fine surgical forceps for 5 s), 30 mM spermidine administrated in drinking water exerted neuroprotective effects through anti-oxidative stress, facilitating optic nerve regeneration and ameliorating apoptosis of RGCs through the inhibition of apoptosis signal-regulating kinase-1 (ASK1)-p38 pathway, which is related to multiple types of stress [99]. It also worked in an autoimmune encephalomyelitis model of multiple sclerosis. The same study group discovered that spermidine (the same treatment as their former research) ameliorated optic nerve demyelination and prevented RGC loss, where its antioxidant effects played the dominant role [98]. Spermidine can also modulate immune cells; for instance, the amount and function of retinal microglia, a kind of macrophage in the nervous system, that secretes inducible nitric oxide synthase (iNOS) may cause RGC death [110,111]. The systemic administration of spermidine effectively suppressed the amount of microglia and iNOS expression in spinal cords of multiple sclerosis and optic nerve injury, as mentioned above, revealing the biofunction of spermidine in modulating immune cells [66,98,99], where such related inflammatory pathways may also play an essential role in photoreceptor degeneration of the retina [112]. In axonal regeneration, spermidine can reverse the inhibition of myelin-associated glycoprotein (MAG) and myelin, which have been recognized as obstacles to spontaneous axonal regeneration in the central nervous system [20]. In vivo, putrescine must be converted to spermidine to overcome MAG inhibition, while the application of spermidine alone was enough to promote optic nerve regeneration in a dose-dependent manner, as mentioned above [20]. In addition, spermidine can inhibit the metalloenzyme carbonic anhydrase, an established pharmacological target for glaucoma, by anchoring to the zinc-coordinated water molecule [113,114]. Among glaucoma patients, inhibiting carbonic anhydrase can effectively alleviate ocular hypertension by reducing the rate of bicarbonate formation, thus decreasing the secretion of the aqueous humor [115] and exhibiting the potential of spermidine in treating glaucoma.

#### 4.4.2. Retina Degeneration Induced by Different Causes

Spermidine has proved to be essential in the retinal development of neonatal animals [116,117,118]. In the neonatal retina, the relatively high concentration of polyamines (spermine, spermidine, and putrescine), and their discrete localization in developing photoreceptor outer segments and ganglion cells, suggest an essential role [116]. The degeneration and death of photoreceptor cells that may involve adjacent cell layers are chrematistic of diseases such as inherited retinal degenerations and AMD [119,120]. Spermidine has proven to be indispensable in the proliferation of cultured RPEs, and the depletion of polyamines (including spermidine) may block the active migration of cultured RPEs [13,121]. Pathologically, the dysfunction or degeneration of RPE cells is crucial in the pathogenesis of AMD that occurs over an extended period [122,123]. When pretreated with 10 µM spermidine for l hour before incubation with H_2_O_2_ in an in vitro experiment to simulating the pathogenesis of AMD, spermidine prevented apoptosis of RPEs (ARPE-19 cells) by acting as a specific [Ca^2+^]i chelator to ameliorate Ca^2+^ overload and promote RPE entry into the cell circle by serving as a restorer of DNA damage [14]. It can also downregulate the expression of apoptotic genes in the apoptosis pathway and upregulate the expression of antioxidant enzymes, thus preventing cell death after H_2_O_2_ application [14]. However, as the dose of spermidine increases it becomes harmful to the RPE cell culture. For example, when incubated with 100–500 μM or 10 mM spermidine, much higher doses than those applied by Kim et al. (10 μM), spermidine or spermine induced oxidative stress and toxic aldehyde during their metabolism that may lead to cell death [108,124].

Similarly, an in vivo study also revealed that higher doses are detrimental to rats. Specifically, when injected into vitreous bodies at the dose of 20 or 30 nmol/eye (10 μL), spermidine damaged the structure and function of the retina [108]. In that study, a 10 nmol/eye injection did not cause any effects; a 20 nmol/eye spermidine injection induced reversible RPEs and photoreceptor cell degeneration (disruption of photoreceptor outer segments), while a 30 nmol/eye injection led to significant necrosis in RPE cells and in photoreceptor cell apoptosis [108]. At such doses, spermidine impaired retinal electrophysiological, and barrier functions where the hyperpermeability of the blood–retinal barrier and ERG wave (both a and b waves) amplitude decreased [108].

Cellular ROS increased when exposed to H_2_O_2_, which may have triggered ER stress, thus stimulating the excessive release of Ca^2+^, which led to RPE cell apoptosis [14]. Applying spermidine in vitro, rescued the RPE cells from oxidative and ER stress by reducing Ca^2+^ overload and caspase-8 activity [14]. It acted as a specific [Ca^2+^]i chelator that blocked the increase of intracellular Ca^2+^ in a ROS-independent manner, protecting RPE cells against H_2_O_2_-induced cellular damage [14]. Although spermidine did not reduce ROS levels in their study, it upregulated the expression of antioxidant enzymes such as Kelch-like ECH-associated protein (Keap1) and heme oxygenase-1 (HO-1) [14]. It also modulated the elevated expression of caspase-8 and death receptor 4 (DR4), caspase-8 overactivity, MMP (∆Ψm) loss, downregulation of Bcl-2, cytochrome c release into the cytoplasm, and caspase-9 activation, which had been altered by H_2_O_2_ and play a vital role in the cell apoptosis pathway [14]. In addition, γH2AX, a sensitive marker for DNA damage that significantly increased after H_2_O_2_ application, was suppressed by spermidine, where it stimulated RPE cells to differentiate and prevent their senescence by repairing the damaged intracellular DNA that stopped cells from entering mitosis due to G2 checkpoint inhibition [14]. Despite its positive role in cell proliferation, some scholars are still concerned that this biofunction might potentially induce the formation of epiretinal membranes and then cause severe vision loss. This deserves further investigation [13].

Concerning toxicity, some scholars said it might be a byproduct of spermidine-generated by serum components, not spermidine itself. In an RPE cell culture, spermidine induced ARPE-19 cell death in the presence of fetal bovine serum in a concentration-dependent manner (100–500 μL) [108]. The application of polyamine oxidation pathway inhibitors such as N-acetylcysteine (NAC) and the metabolizing enzyme aldehyde dehydrogenase (ALDH) rescued ARPE-19 cell viability, indicating that excessive serum amine oxidase, oxidative stress, and aldehyde generated during spermidine oxidation may be involved in spermidine-induced cell death [106]. A similar mechanism of inducing damage 24 h after administration in ovaria of female rats was also confirmed when the dose was raised to 150 mg/kg (intraperitoneal), a supraphysiological dose [125].

#### 4.4.3. Altered Level of Spermidine in Proliferative Disorders of the Retina

The association between elevated levels of spermidine and diabetic retinopathy has been elucidated and can be observed in serum, erythrocyte, and vitreous samples [126,127,128]. In serum, significantly higher concentrations of spermidine were observed in both background diabetic retinopathy (0.37 ± 0.05 μg·L^−1^) and proliferative diabetic retinopathy (0.39 ± 0.09 μg·L^−1^) compared to diabetic patients without retinopathy (0.34 ± 0.08 μg·L^−1^). No significant difference was found between background and proliferative diabetic retinopathy groups [126]. In erythrocytes, spermidine was significantly higher among PDR patients (20 ± 7 nmol/mL PRBCs) than in healthy patients (14.5 ± 4 nmol/mL PRBCs) [127]. Moreover, in human vitreous samples, spermidine levels were three times higher in quiescent PDR and about four times higher in active PDR and proliferative vitreoretinopathy (PVR) compared with those in the control group, indicating that elevated spermidine may be associated with PDR and PVR progression [128]. Among the active, but not quiescent, PDR and PVR groups, a positive correlation was found between higher spermidine levels and two elevated angiogenic factors, VEGF and IL-8, which is connected to active vascular proliferation epiretinal membrane formation; however, the mechanism remains to be elucidated [128]. In addition, a Chinese study group developed a novel PDR rat model via the intravitreal injection of 10 μL spermidine at a dose of 20 nmol/eye in streptozocin (STZ)-induced diabetic rats to induce oxidative stress under high glucose conditions [129]. They successfully observed significantly higher ROS and much more severe histopathological changes in the STZ^+^ spermidine group than in the STZ and healthy groups.

Retinopathy of prematurity (ROP), another vasoproliferative disease in developing retinal vessels, may lead to severe and irreversible vision loss if not treated properly [130]. Gralek et al. supposed that hyperoxia, known to cause retardation of vessel growth, was related to reduced enzyme activity and polyamine levels in newborn rats for the first 2 weeks of life. They exposed newborn rats to hyperoxia for two weeks and then assessed the levels of polyamines (including spermidine) and enzyme activity (such as ODC and SSAT) in the retina and lens: they both decreased [131]. In another study, Gralek et al. tried to establish a murine model of ROP but failed, but a similar significant variation was not found [132].

These findings suggest that spermidine may be a marker for proliferative disorders of the retina, and its role in PDR, PVR, and ROP remains to be elucidated.

## 5. Conclusions and Open Questions for Further Study

Polyamine research, especially concerning spermidine, has shown considerable progress over the past years, exhibiting its potential to be a therapeutic agent for various diseases in different systems [23,31,45,51,133,134]. To our knowledge, this is the first review of spermidine’s application in ocular diseases. We identified its role and delineated its potential as a therapeutic agent or novel biomarker for glaucoma, optic nerve injury, AMD, cataracts, dry eye syndrome, bacterial keratitis, KC, PDR, PVR, and ROP (summarized in Table 1). The detrimental and therapeutic effects of spermidine are summarized in Figure 1 and Figure 2.

Given the significant influence that spermidine can exert on other cells and tissues, such as modulation of the immune microenvironment, acceleration of neuro-regeneration, and repair of barriers, we have more expectations for its further application. Compared with earlier studies that focused mainly on the application and mechanism of spermidine, we also noticed different voices that may be explained by the different doses and routes of administration. Therefore, it is worth mentioning that these should be scrutinized for treating ocular diseases in case they should induce excessive ROS, as we summarized in Table 2. In addition, when considering further applications of spermidine in ocular diseases, we should also take potential risks into account, including the progression of PDR, excessive migration and proliferation of RPE cells, and accelerating inflammation in bacterial keratitis in the absence of antibiotics.

## Figures and Tables

**Figure 1 pharmaceutics-14-01500-f001:**
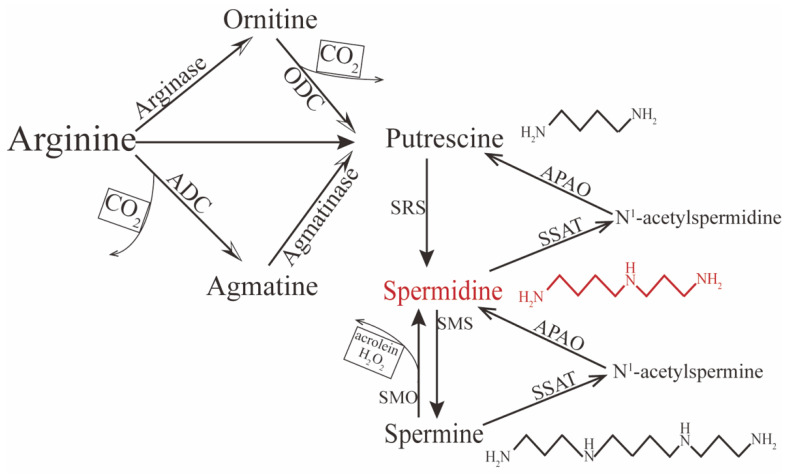
Polyamine’s synthesis and catabolism in mammals.

**Figure 2 pharmaceutics-14-01500-f002:**
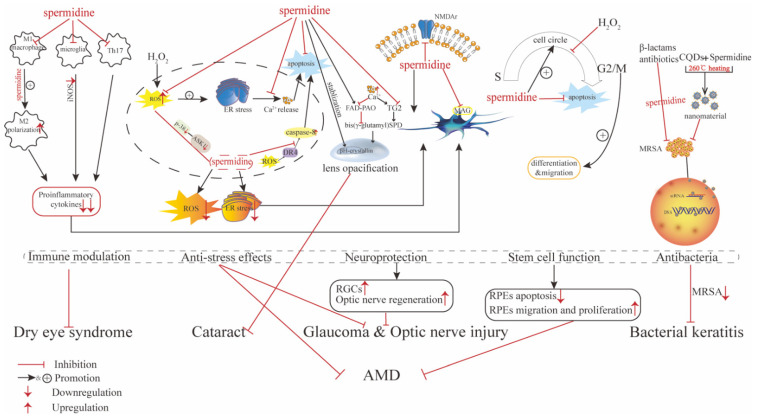
Protective effects and the mechanism that spermidine exerts on ocular diseases.

**Table 1 pharmaceutics-14-01500-t001:** A summary of altered levels of spermidine in ocular diseases.

Diseases	Spermidine Variation	Samples	References
Glaucoma	decreased	Serum, plasma, and aqueous humor (human)	[91,92,93]
Cataract	decreased	Lens (rabbit)	[19,82]
KC	decreased	Corneal fibroblasts (human)	[78]
Uveitis	Decreased in 48 h then increased	Aqueous humor (rabbit)	[86,87]
PDR	increased	Serum, erythrocyte, and vitreous (human)	[124,125,126]
PVR	increased	Vitreous (human)	[126]
ROP	Decreased (first 14 days)	Retina and lens (rat)	[129]

KC: keratoconus; PDR: proliferative diabetic retinopathy; PVR: proliferative vitreoretinopathy; ROP: retinopathy of prematurity.

**Table 2 pharmaceutics-14-01500-t002:** A comparison between different routes and doses administration.

Study Type	Dose (Route)	Task	Findings	References
in vitro	1–30 μM (cc)	Reverse apoptosis of RPEs treated with H_2_O_2_	Blocked the increase of intracellular Ca^2+^, exhibiting anti-oxidative effect. Concentrations over 20 μM were found to be toxic.	[14]
in vitro	10 μM (cc)	Assess RPEs migration	Promoted the migration of RPEs.	[13]
in vitro	10 μM (cc)	Assess its role in RPEs proliferation	Being indispensable in RPEs proliferation.	[119]
in vitro	10 mM (cc)	Assess cytotoxic effect of spermine and spermidine on RPE cells	10mM spermidine caused slight RPEs shrinkage and reduced cell density; while spermine exhibited significant cytotoxicity.	[122]
in vitro	100 μM (cc)	Counter the effect of NMDAr inhibitor in retinal neuronal cells	Reversed the neuro-protective effect of nafamostat, a NMDAr inhibitor, in the cell viability analysis.	[105]
in vitro	100–500 μM (cc)	Mediate oxidative degeneration of RPEs	Induced RPEs death in vitro.	[106]
in vivo	10 μL, 30 nmol/eye (i.v.i)		Induced RPEs and photoreceptor cell degeneration in rats.	
in vivo	10 μL, 20 nmol/eye (i.v.i)		Induced reversible RPEs and photoreceptor cell degeneration in rats.	
in vivo	10 μL, 10 nmol/eye (i.v.i)		No significant harmful effects were witnessed.	
in vivo	5 µL, 50 nmol/eye (i.v.i)	Counter the effect of NMDAr inhibitor in retina	Reversed the neuro-protective effect of nafamostat, a NMDAr inhibitor, after the application of neurotoxic NMDA.	[104]
in vivo	10 μL, 20 nmol/eye (i.v.i)	Establishment of PDR murine model	Contributed to the establishment of PDR murine model.	[127]
in vivo	10 μL, 20 pmol/eye (i.v.i)	Assess optic nerve regeneration	Promoted optic nerve regeneration after injury.	[20]
	10 μL, 40 pmol/eye (i.v.i)		No significant effects were observed.	
in vitro	20 μM (cc)	Assess the dose and the effect of overcoming inhibition by MAG and myelin	Could effectively reverse the inhibition of MAG and myelin.	
	100 μM (cc)		Diminished effects.	
	500 μM (cc)		No effects.	
in vivo	10 mg/kg (s.c.)	Prevent monosodium glutamate neurotoxicity in the rat retina	Prevented neurodegeneration and neurotoxicity of monosodium glutamate in retina.	[98]
	1 mg/kg (s.c.)		No effects.	
in vitro	20 µL, 0.2/0.5% (Gutt)	Ameliorate dry eye syndrome induced by PM_2.5_	Alleviated the symptom of dry eye murine model. Exhibited anti-inflammatory effects on ocular surface.	[17]
in vivo	30 mM in drinking water (p.o.)	Assess its protective role in optic neuritis and retinal neurons.	Alleviated optic neuritis and improved visual function.	[96]
in vitro	4 mM (cc)		Protected RGCs from H_2_O_2_ induced oxidative stress.	
in vivo	30 mM in drinking water (p.o.)	Investigate its protective role after optic nerve injury	Prevented RGCs death, suppressed retinal degeneration, and enhanced optic nerve regeneration.	[97]
in vivo	30 mM in drinking water (p.o.)	Investigate its protective role in normal tension glaucoma murine model	Suppressed retinal degeneration, ameliorated visual impairment, and reduced oxidative stress level in retina.	[16]
in vitro	100 mM (tissue incubation)	Assess its role in delaying lens opacification.	Ameliorate lens opacification induced by Ca^2+^.	[19,82]

s.c.: subcutaneous injection; p.o.: oral; i.v.i: intravitreal injection; Gutt: eye drop; cc: cell culture.

## Data Availability

Not applicable.

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
