# Peer review of "Research Progress and Potential Applications of Spermidine in Ocular Diseases"

_pharmaceutics, 2022, doi:10.3390/pharmaceutics14071500_

Round 1

Reviewer 1 Report

Name of journal: Pharmaceutics

Manuscript ID: pharmaceutics-1770835

Title: Research progress and potential applications of spermidine in ocular diseases.

The " Research progress and potential applications of spermidine in ocular diseases" by Han et al. is overall a well-written review article. The authors describe the applications of spermidine in several ocular diseases.

Please find attached some minor corrections that could be implemented to improve the fruition of the present review article.

1)     Expand the introduction by mentioning eye disorders and their impact on public health

2)     Write the names of the bacteria in italics

3)     You have reported the use of spermidine for the treatment of bacterial keratitis. It would be advisable to report the mode of action of the spermidine.

4)     Report sperm applications in viral eye infections

5)     It seems only right to indicate the effects of sperm on the ocular microbiota. Cite the article: "Current Evidence on the Ocular Surface Microbiota and Related Diseases" (PMID: 32668575, DOI: 10.3390 / microorganisms8071033)

6)     Double-check the abbreviations throughout the text.

Kind regards

Author Response

1. The reviewer’s comment: Expand the introduction by mentioning eye disorders and their impact on public health.

The authors’ answer: We appreciate very much the positive comments and suggestions of the reviewer on the previous version of our manuscript. In agreement with these valuable suggestions and recommendations, we carefully revised our manuscript. According to your comment, we have expanded the introduction of ocular disorders and their impacts on public health to improve this review article.

2. The reviewer’s comment: Write the names of the bacteria in italics.

The authors’ answer: Thanks for your suggestion, we have checked and corrected the writing forms of microorganisms in the manuscript thoroughly.

3.The reviewer’s comment: You have reported the use of spermidine for the treatment of bacterial keratitis. It would be advisable to report the mode of action of the spermidine.

The authors’ answer: Thanks for your comment, in the section 4.1.2, we supplemented the mode of action of spermidine in bacteria and updated the related mechanism in Figure 2.

4. The reviewer’s comment: Report sperm applications in viral eye infections

The authors’ answer: Thanks for your profound suggestion. We have searched the databases but did not find out the direct application of spermidine in viral eye infections. Interestingly, compound-driven (e.g., spermidine) induction of autophagy limits SARS-CoV-2 propagation (PMID: 34155207). When synthesized into persulfate molecular umbrellas, spermidine can also exert anti-HIV and anti-HSV properties (PMID: 15584704). Spermidine’s application in viral eye infections deserve further research.

5. The reviewer’s comment:  It seems only right to indicate the effects of sperm on the ocular microbiota. Cite the article: "Current Evidence on the Ocular Surface Microbiota and Related Diseases" (PMID: 32668575, DOI: 10.3390 / microorganisms8071033)

The authors’ answer: Thanks for the valuable suggestion, which gave us a profound understanding of ocular microbiota and infection. We have read and cited the article "Current Evidence on the Ocular Surface Microbiota and Related Diseases" in section 4.1.2, line 370-372, discussing the potential role of spermidine in modulating gut microbiota whose balance is in close connection with ocular microbiota.

6. The reviewer’s comment:  Double-check the abbreviations throughout the text.

The authors’ answer: Thanks for your suggestion, we have conducted typo corrections carefully to minimize typewritten, grammatical, and bibliographical errors under your guidance.

Reviewer 2 Report

Han et al. realized a very interesting review describing the “Research progress and potential applications of spermidine in ocular diseases”. I consider the manuscript very fascinating but, at the same time, I suggest several revisions needed to improve the reliability and the completeness of the paper: 

·      Sections 4.4.2 and 4.4.3. The “Inflammation” and “Oxidative Stress” molecular mechanisms correlated with retinal diseases were not sufficiently described. Thus, I suggest the authors to add the recent PMID: 34440511, PMID: 34058230 and PMID: 33801777, which could represent a substrate able to enforce the role of considered cellular mechanisms.

·      Finally, manuscript requires English revisions and typos correction. 

Author Response

The reviewers comment:

Sections 4.4.2 and 4.4.3. The “Inflammation” and “Oxidative Stress” molecular mechanisms correlated with retinal diseases were not sufficiently described. Thus, I suggest the authors to add the recent PMID: 34440511, PMID: 34058230 and PMID: 33801777, which could represent a substrate able to enforce the role of considered cellular mechanisms.

Finally, manuscript requires English revisions and typos correction. 

The authors’ answer: As we conquer with reviewer comment, we added more associated articles about inflammation and oxidative stress in retinal degeneration (PMID: 34440511 in section 4.4.1, page 7, line 762-764, [100]; PMID: 34058230 in section 4.4.1, page 7, line 802-803, [110]; PMID: 33801777 in section 4.4.2, page 8, line 890-892). Finally, we conducted English revisions and typos corrections carefully to minimize typographical, grammatical, and bibliographical errors.

Reviewer 3 Report

Comments to Author:

Section 2. Metabolism of Spermidine

Page 2, line 51: units is missing - molecular weight of 145.25??? (unit).

Page 2, line 58: add space between the value and the unit (e.g. 5-15 mg/day). (Check through entire text.) See also table 2.

Section 3. Biofunctions of Spermidine

The first paragraph is pointless, superfluous

Page 3, line 106: Ca2+ should be written as Ca2+ (superscript). (Check through entire text.)

Section 4.1.2. Bacterial keratitis

Page 4, line 167: Latin names of microorganisms are written in italics (e.g. Staphylococcus aureus) (Check through entire text.)

Section 4.4.3. Altered level of spermidine in proliferative disorders of the retina

Page 9, line 415: please check the unit for concentration ??? -  ρ/μg·L-1

Figures and tables

Figure 1: Check subscript of numbers in formulas (e.g. NH2 etc.)

Table 1: The name of column Variation should be changed to Level of spermidine.

Section Conclusion

The conclusion should be more impressive. In addition, the authors should highlight their innovations compared to similar review articles. What we can read in their review article that other similar review articles do not cover.

Author Response

The reviewers comment:

Section 2. Metabolism of Spermidine

Page 2, line 51: units is missing - molecular weight of 145.25??? (unit). 

Page 2, line 58: add space between the value and the unit (e.g. 5-15 mg/day). (Check through entire text.) See also table 2.

Section 3. Biofunctions of Spermidine

The first paragraph is pointless, superfluous

Page 3, line 106: Ca2+ should be written as Ca2+ (superscript). (Check through entire text.)

Section 4.1.2. Bacterial keratitis

Page 4, line 167: Latin names of microorganisms are written in italics (e.g. Staphylococcus aureus) (Check through entire text.)

Section 4.4.3. Altered level of spermidine in proliferative disorders of the retina

Page 9, line 415: please check the unit for concentration ??? -  ρ/μg·L-1

Figures and tables

Figure 1: Check subscript of numbers in formulas (e.g. NHetc.)

Table 1: The name of column Variation should be changed to Level of spermidine.

The authors’ answer: Thanks for your valuable comments, we have revised the points according to your comments. We conducted typo corrections carefully to minimize typewritten, grammatical, and bibliographical errors under your guidance.

Section Conclusion

The conclusion should be more impressive. In addition, the authors should highlight their innovations compared to similar review articles. What we can read in their review article that other similar review articles do not cover.

The authors’ answer: Thanks for your suggestion, we have highlighted our innovations compared with other similar review articles that focus on the effects of spermidine.